# Clinical Course and Mutational Analysis of Patients with Cystine Stone: A Single-Center Experience

**DOI:** 10.3390/biomedicines11102747

**Published:** 2023-10-11

**Authors:** Jae Yong Jeong, Kyung Jin Oh, Jun Seok Sohn, Dae Young Jun, Jae Il Shin, Keum Hwa Lee, Joo Yong Lee

**Affiliations:** 1Department of Urology, National Health Insurance Service Ilsan Hospital, Goyang 10444, Republic of Korea; urojjy@nhimc.or.kr; 2Department of Urology, Chonnam National University Hospital, Chonnam National University Medical School, Gwangju 61469, Republic of Korea; exeokj@hanmail.net; 3Department of Medicine, Yonsei University College of Medicine, Seoul 03722, Republic of Korea; junseok.sohn19@med.yuhs.ac; 4Department of Urology, Severance Hospital, Urological Science Institute, Yonsei University College of Medicine, Seoul 03722, Republic of Korea; dyjun881101@yuhs.ac; 5Department of Pediatrics, Yonsei University College of Medicine, Seoul 03722, Republic of Korea; shinji@yuhs.ac; 6Division of Pediatric Nephrology, Severance Children’s Hospital, Seoul 03722, Republic of Korea; 7Institute of Kidney Disease Research, Yonsei University College of Medicine, Seoul 03722, Republic of Korea; 8Center of Evidence Based Medicine, Institute of Convergence Science, Yonsei University, Seoul 03722, Republic of Korea

**Keywords:** cystinuria, mutation, urolithiasis

## Abstract

Cystinuria is a known genetic disorder. To date, two genes, SLC3A1 and SLC7A9, have been identified as causes of cystinuria. In this study of 10 patients with cystinuria, which is the largest Korean cohort ever studied, we examined the patients’ phenotypes, clinical courses, and genetic analyses. A total of 10 patients with cystinuria diagnosed with cystine stones in a single tertiary medical center (Severance Hospital, Seoul, Republic of Korea) from April 2000 to July 2023 were included in the study. All of the patients participated in mutational studies, and the clinical presentation and consecutive laboratory findings of the patients were analyzed retrospectively. After the initial stone-related surgery or procedure at our hospital, 6 of the 10 patients underwent additional surgery at least once for recurrent stones. Genetic analyses identified six new mutations, of which only two patients had type B mutations. The most common genotype was compound heterozygous type A. We investigated the genotypes and clinical courses of 10 Korean patients with cystinuria who had not been previously reported. More data are needed to statistically analyze the genotype and phenotype of cystinuria.

## 1. Introduction

Cystinuria is a known genetic disorder that causes abnormalities in the renal transepithelial transporter, which in turn interferes with the reabsorption of dibasic amino acids (including cystine, ornithine, lysine, and arginine) and results in urinary hyperexcretion. Of these four amino acids, only cystine has low solubility at normal urinary pH and precipitates to form cystine stones [1,2].

To date, two genes have been identified to cause cystinuria: SLC3A1 and SLC7A9. Located on 2P16.3, SLC3A1 encodes the heavy subunit of the renal amino acid transporter (rBAT). Patients with mutations in this gene are classified as having type A cystinuria. On the other hand, SLC7A9, located on 19q13.1, encodes the light subunit of the renal amino acid transporter (b0,+AT). Patients with mutations in this gene are classified as having type B cystinuria. The rBAT/b0,+AT transporter is linked by a disulfide bond, and the complex is presented on the apical membrane of proximal tubules in the kidney. Patients with mutations in both genes are classified as having type AB cystinuria [3,4].

Although cystinuria is the most commonly inherited form of kidney stone disease, only 1% of all kidney stone disease cases are due to this disorder. Cystine stones account for 1–2% of urolithiasis in adults but 10% in children, and there are ethnic differences in worldwide prevalence [5,6]. Global prevalence is estimated to be 1 in 7000, and this value ranges from 1 in 100,000 (Swedish) to 1 in 2500 (Libyan). According to an analysis of the composition of kidney stones in a large database of Korean patients with kidney stones, cystine stones accounted for only 0.35% of the total [7]. As such, cystine stones are relatively rare, but with a recurrence rate of up to 60% and stone formation at an early age, cystinuria is a condition that requires lifelong management as it can reduce the patient’s quality of life and lead to chronic kidney disease (CKD).

As genetic analysis has become increasingly common, various papers have been published including a mutational analysis study of cystinuria patients, and some genes characteristic of different regions have been reported. However, to date, only four papers have reported on genetic analysis in Koreans, with a total of 18 patients [8,9,10,11]. In this study, we assessed the genetic analysis, phenotype, and clinical courses of 10 novel cystinuria patients which is the largest Korean cohort studied in this manner. Also, we compared the phenotypes of patients with type A and type B cystinuria to determine the association of genotype with the clinical course of cystinuria. Then, we checked the geographical characteristics of Korean cystinuria mutation by adding our data to the existing Korean patients.

## 2. Materials and Methods

### 2.1. Patients

We searched the database for stone composition analysis results and selected all cases reported as cystine stones without exception. A total of 10 cystinuria patients diagnosed with cystine stones from April 2000 to July 2023 in a single tertiary medical center (Severance Hospital, Seoul, Republic of Korea) were included in the study. After the cystine stones were confirmed, all patients were subject to mutational analysis and were followed up regularly to check renal function and stone recurrence, with various laboratory tests, including the estimated glomerular filtration rate (eGFR) calculated using the Schwartz formula or the Chronic Kidney Disease Epidemiology Collaboration (CKD-EPI) equation according to age and low-dose non-contrast computed tomography (CT) at 6-month intervals. Patients’ age at onset and diagnosis, sex, familial history, recurrent stone events, laboratory findings, urinary amino acid excretion, clinical symptoms, the usage of alkalinizing agents or thiol-binding agents, and mutational analysis were reviewed retrospectively.

### 2.2. Surgical Treatment, Stone Analysis, and Measurement of Urinary Amino Acids

All patients had at least one intervention for stone removal. Some patients had previously undergone open nephrolithotomy at other hospitals. All but one patient with laparoscopic ureterolithotomy at our institution underwent retrograde intrarenal surgery (RIRS) or endoscopic combined intrarenal surgery (ECIRS) given the size and location of the stones [12,13]. Lithotripsy was performed with a holmium:YAG laser lithotripter (VersaPulse^TM^ PowerSuite^TM^ 100 W; Lumenis, Tel Aviv, Israel). The extraction of fragmented stones was carried out with stone basket for RIRS and a passive washout mechanism for ECIRS. During the follow-up, additional surgery was performed if the migration of the renal stone resulted in an event that caused symptoms such as renal colic and obstructive pyelonephritis, or if the stone continued to increase in size and showed a staghorn pattern on the CT scan.

Extracted stone particles were collected and sent to GC Laboratories (Yongin, Republic of Korea) for the quantitative analysis of stone composition using Fourier transform infrared spectroscopy. Random urine samples from patients with confirmed cystine stones were obtained and sent to Seoul Clinical Laboratory (Yongin, Republic of Korea) for the quantitative analysis of excreted amino acids using ion exchange chromatography.

### 2.3. Mutational Studies

Genomic DNA samples extracted from the peripheral blood lymphocytes of each patient were used for library preparation and target capture with a custom panel that targeted candidate genes. Massively parallel sequencing was performed on the NextSeq 550Dx System (Illumina; San Diego, CA, USA). GRCg37 (hg 19) was used as the reference sequence for mapping and variant calling. Databases used for analysis and variant annotation include the Online Mendelian Inheritance in Man (OMIM), the Human Gene Mutation Database (HGMD), ClinVar, dbSNP, the 1000 Genome Project, the Exome Aggregation Consortium (ExAC), the Exome Sequencing Project (ESP), and the Korean Reference Genome Database (KRGDB). The classification of variants followed the standards and guidelines established by the American College of Medical Genetics (ACMG). All pathogenic and likely pathogenic variants were further confirmed by Sanger sequencing.

### 2.4. Statistical Analyses

The entire patient cohort was divided into two groups according to their genotype, (the type A cystinuria group and the type B cystinuria group) in order to compare the phenotypes. The age at onset (years), sex, the presence of bilateral stones, urinary cystine excretion, the number of interventions, and current eGFR inhibitors were set as variables. Normality was tested for each variable using the Shapiro–Wilk test. Data were expressed as mean ± standard deviation if normally distributed and as the median (range) otherwise. Considering the sample size, the Mann–Whitney U test was used for statistical comparisons of continuous variables. Fisher’s exact test was used to compare categorical variables. Differences were considered statistically significant when the *p* value was less than 0.05. All computations were performed using the R version 4.3.0 (R Foundation for Statistical Computing, Vienna, Austria; http://www.r-project.org).

### 2.5. Ethics Statement

This study was approved by the Institutional Review Board at Severance Hospital, Yonsei University Health System, Seoul, Republic of Korea (approval no. 4-2023-0008; approval date: 1 March 2023). However, the requirement for the written informed consent of subjects was waived because this is a retrospective analysis of clinical data obtained while treatment that has already ended and does not involve more than minimal risk to the subject and all patient records and data were anonymized in advance to strictly maintain the confidentiality of the personal information of the subjects throughout the entire research process.

## 3. Results

All patients underwent at least one stone-related intervention, including laparoscopic ureterolithotomy and open nephrolithotomy. Six of the ten patients were male and four were female. The median age at onset was 16 years and the median age at diagnosis was 19 years (ranges: 0.8–62.5 and 8.8–64.0, respectively). Six of the ten patients were diagnosed with stones for the first time at pediatric age. The mean follow-up period was 7.1 years, with a median of 6.1 years. The initial symptoms included urinary tract infection (n = 3) and hematuria with proteinuria (n = 1). The remaining six patients were diagnosed with stones due to stone-related pain or chance. Two patients reported a family history of stones when questioned. Case 3 reported that his maternal grandmother underwent nephrectomy due to urolithiasis and Case 10 reported a family history of urolithiasis in both the grandfather and father. Four patients had bilateral stones at the time of initial intervention. All except two patients had urolithiasis on their last follow-up non-contrast CT imaging of the kidney, ureters, and bladder. After the initial stone-related surgery or procedure at our hospital, six of the ten patients underwent additional surgery for recurrent stones at least once. In terms of renal function, four patients currently had an eGFR equivalent to CKD stage 2, whereas all others were in CKD stage 1. One of those patients had an eGFR of 48 mL/min/1.73 m^2^ on initial treatment, which rose to 96.3 mL/min/1.73 m^2^. Three patients were taking tiopronin (thiola^®^) daily (Table 1).

Eight patients underwent 24 h urinary excretion tests, and all showed urinary cystine excretion above the normal range. All dibasic amino acids, other than cysteine (ornithine, lysine, and arginine), were also elevated (Table 2).

Genetic analysis identified six new mutations, with only two patients harboring type B mutations. The most common genotype was compound heterozygous type A (4/10) (Table 3).

When patients were divided into two groups (the type A cystinuria group and the type B cystinuria group), no statistical correlation was found in terms of sex, the age of onset, bilateral stone presentation, the number of interventions, or current eGFR (Table 4).

Mutational analysis was available for a total of 18 Korean patients from three previously published studies (Table 5). Following the mutational analysis of the 10 patients in this study, a total of 24 mutations were identified in Korean patients (Figure 1).

## 4. Discussion

The genotype and phenotype of the largest cohort of cystinuria patients from a single institution in Korea are summarized in this study. Six novel mutations were identified and considered to be pathogenic based on their frequency in the normal population and the expected high pathogenicity of in silico analysis.

The clinical presentation of cystinuria varies in the literature. For example, the male-to-female ratio is reported to be 3:2, which differs from the previously reported Korean cohorts [10,11]. Dello Strologo et al. reported that male patients had a more severe phenotype than female patients [14]. However, our data showed a relatively more repetitive intervention rate in female patients (median: male = 3.5 vs. female = 4.0). Although there is variation in the literature, less than 80% of patients reportedly have their first stones within the first two decades of life, and 1–16% of patients present with the first stone before age 40, which closely recapitulates the results in our cohort [15,16].

Unlike calcium-based stone formers, for which up to 25% reportedly have bilateral stone formation, cystinuria is characterized by up to 75% bilateral stone formation. However, no significant difference in renal function between bilateral and unilateral stone formers has been reported, even though bilateral stone formers require relatively more intervention than unilateral stone formers [17]. In this study, four patients had bilateral stones at the time of initial intervention and these patients had more additional intervention than those who had unilateral stones (5.5 ± 2.4 vs. 3.3 ± 3.6, *p* = 0.326). Nevertheless, current renal function with eGFR was similar between the two groups (95.1 ± 18.6 vs. 88.9 ± 21.2, *p* = 0.646).

Although there is an increased risk of CKD in patients with cystinuria, actual kidney failure is known to be less than 5% [18], and our patients presented with at least CKD stage 2 at their last follow-up evaluation.

Prot-Bertoye et al. showed that CKD progression in cystinuria patients is associated with age, hypertension, and severe renal parenchyma damage [19]. Therefore, one important component of treating cystinuria is to prevent an increase in stone burden, thereby reducing the number of urological interventions and damage to renal parenchyma [20,21]. At our institution, tiopronin was prescribed first if the patient had a high stone burden and frequent recurrences. Unfortunately, due to production and supply shortages, only three patients are currently being treated with tiopronin (Cases 5, 7, and 9). Of the patients who attempted tiopronin therapy, two patients discontinued use due to adverse drug reactions. Case 4 developed OT/PT elevations and Case 6 discontinued use due to allergic symptoms including fever, colitis, and presumed drug eruption. Two additional patients (Case 1 and Case 3) had no adverse events but were discontinued because the drug was no longer available and were given a plan to receive it when it becomes available. If medication was needed and tiopronin was not available, medications such as potassium citrate or D-penicillamine (Case 1 and Case 8) were considered [22]. Captopril, which is a thiol first-generation ACE inhibitor, was also prescribed (Case 1 and Case 9). Although it is currently an off-label administration, there have been published studies showing effectiveness in cystinuria because captopril has a sulfhydryl ligand that bonds with cystine and forms cystine–captopril disulfide, which is more soluble than cystine [23,24]. No significant adverse effects were reported when non-tiopronin drugs were administered, and the addition of tiopronin is being considered based on the availability of the drug and the recurrence of urolithiasis.

Regarding the genotypes of cystinuria, type A cystinuria was generally considered as an autosomal recessive disease; however, subsequent studies have identified cases of high urinary cystine excretion and stone formation in heterozygous individuals [3,25]. In this study, two patients with heterozygous type A mutation showed above-normal cystinuria levels. The type B mutation is known to be autosomal-dominant with incomplete penetrance. Consistent with this knowledge, two type B patients in our cohort showed a heterozygous genotype without familial history.

Of the 12 mutations reported, the most common were missense mutations, followed by 2 frameshifts and 1 deletion. Wong et al. hypothesized that patients with missense mutations have less severe disease because missense mutations result in a change to a single amino acid within the protein compared with all other types of mutations that cause larger genomic alterations [26]. In their study, patients with at least one missense mutation in *SLC3A1* had significantly lower levels of urinary lysine, arginine, and ornithine, but not cystine, than patients with any other combination of mutations. In contrast, our data showed no significant difference in amino acid excretion levels in patients with missense mutations (cystine, *p* = 0.945; ornithine, *p* = 0.402; lysine, *p* = 0.991; and arginine, *p* = 0.537).

Geographical variations in cystinuria mutations have been reported. For example, p.T216M in SLC3A1 and p.G105R in SLC7A9 are common in Greece and dupE5E9 is more commonly found in Germany. In the Asian population, SLC7A9 c.1445C>T (p.P482L) is common and might indicate a mutation hotspot in the Japanese population [6,27,28]. Among the total of 24 identified mutations, the aforementioned common mutations p.T216M and p.P482L were also found. However, the common mutations in the Korean population were c.1820delT, c.1976A>C, and c.2017T>C in SLC3A1 (28.5% in total).

We compared the clinical presentations between type A and type B cystinuria and observed no significant differences. These results are consistent with previous studies [10,14,26,29]. Therefore, current research suggests that there is no difference in the clinical course and characteristics of cystinuria based on genotype, and both types require ongoing renal function testing and continued follow-up for stone recurrence. However, these results do not mean that genetic testing for cystinuria is not necessary. For example, studies have been published showing no pathogenic variants in SLC3A1 or SLC7A9 in 5~10% of patients with confirmed cystine stones [16,30]. Wu et al. reported discrepancy between clinical and genetic prevalence (1 in 7000 individuals vs. 1 in 30,585 estimates) and suggested novel causal genes, undiscovered pathogenic variants, alternative inheritance models, founder effects, epigenetic modification, environmental factors, modifier genes, or other modification factors as possible causes [31]. In our cohort, family history was identified in only 2 out of 10 patients, and 2 patients (Case 1 and Case 6) showed a heterozygous *SLC3A1* mutation with no family history, indicating autosomal dominance with incomplete penetrance rather than autosomal recessiveness. In this regard, previous studies suggested the existence of genetic modifiers or novel cystine transporters [32,33]. Given these considerations, genetic analysis still has an important role in helping us understand the pathogenesis of cystinuria.

Furthermore, while research on cystinuria has primarily focused on clinical and genetic aspects to date, considering patient groups without *SLC3A1*/*SLC7A9* mutations, it appears that additional studies are necessary, including research on the physicochemical factors related to cystine aggregation, similar to the results observed in calcium stone studies, and investigations into other urinary tract symptoms that may arise due to cystine crystal formation [34,35].

In summary, this study followed the clinical course of 10 Korean patients with cystinuria, and, despite recurrent stones, the patients’ renal function was well maintained with continuous follow-up. Although no statistical association between genotype and phenotype was found, mutation analysis showed that *SLC3A1* mutation was more frequent in Korean patients, in contrast to previous analyses in Europeans and Japanese.

This study had some limitations. First, the 10 patients in this study comprised a relatively small sample size. Only two type B mutations were identified in this cohort, and these two patients were not available for urinary amino acid excretion testing. More patients should be enrolled in future studies to more precisely compare the clinical course according to mutation. Second, the study relied only on patient statements to investigate family history, and did not obtain genetic testing or laboratory data from family members. Such results would have provided additional information on the inheritance pattern and clinical course of cystinuria.

## 5. Conclusions

We investigated genotypes and clinical courses of 10 Korean patients with cystinuria who were not previously reported. No significant association was identified between type A and type B cystinuria on clinical course, including the recurrence of stones. Mutations in the *SLC3A1* gene were more frequently observed in the Korean population.

## Figures and Tables

**Figure 1 biomedicines-11-02747-f001:**
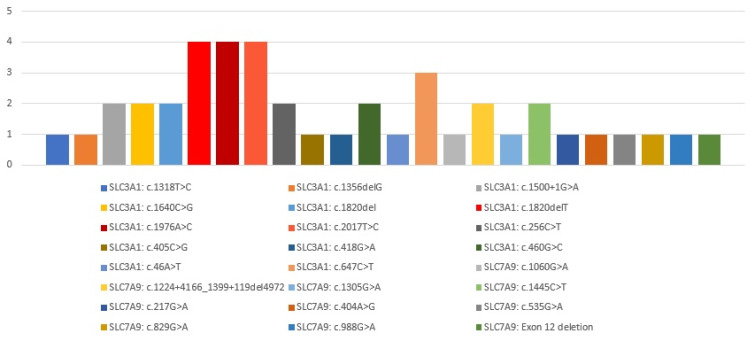
Korean mutational spectrum of *SLC3A1* and *SLC7A9* genes.

**Table 1 biomedicines-11-02747-t001:** Clinical features of 10 patients with cystinuria.

Case	Sex	Age (Years)	Onset (Years)	Diagnosis (Years)	Family History	Initial Symptom	Initial Bilateral Stones	Stone Recurrence *	Initial eGFR	Current eGFR	Medication Other than Tiopronin	Type of Intervention
1	M	38.7	15.0	15.7	-	Stone	+	+	86	97.7	KC, PD, Capto, Artamin	PNL, RIRS
2	M	17.4	0.9	13.8	-	UTI	-	+	63.6	70.0	KC, Pyridoxine	OpenNL, URSL
3	M	10.4	0.8	8.8	+	UTI	-	+	78.0	72.7	KC, PD	ECIRS
4	F	19.7	1.2	13.8	-	UTI	+	+	111.5	69.4	KC, PD, Enala	PCNL, open NL, ECIRS
5	M	20.6	16.0	16.2	-	HU, PU	-	+	107.0	108.2	KC, PD	ECIRS
6	M	27.4	18.6	20.8	-	Stone	-	+	48	96.3	PD	LapaUL
7	F	66.7	62.5	64.0	-	Stone	-	-	62	68.9	KC, PD	ECIRS
8	F	35.1	26.0	33.2	-	Stone	+	+	64	99.4	KC, PD, Artamin	ECIRS
9	M	38.1	16.0	34.1	-	Stone	-	+	90	117.2	KC, Capto, PD, Artamin	OpenNL, RIRS
10	F	28.7	26.8	28.3	+	Stone	+	-	107	114	none	ECIRS, RIRS

HU = hematuria, PU = proteinuria, eGFR = estimated glomerular filtration rate (calculated using the Schwartz formula or CKD-EPI according to age) (mL/min/1.73 m^2^), PD = pyridoxine (vitamin B6), KC = potassium citrate, Capto = captopril, Enala = enalapril, PNL = percutaneous nephrolithotomy, RIRS = retrograde intrarenal surgery, OpenNL = open nephrolithotomy, URSL = ureterorenoscopic lithotomy, ECIRS = endoscopic combined intrarenal surgery, LapaUL = laparoscopic ureterolithotomy. * On the last follow-up imaging test (non-contrast CT or KUB).

**Table 2 biomedicines-11-02747-t002:** Urinary excretion levels (nmol/mg creatinine) of the branched amino acids of the patients.

Case	Cystine	Ornithine	Lysine	Arginine
1	2099.3	930.9	5850.7	3505.1
2	1339.6	1447.2	2942.8	3731.4
3	2111.4	1699.8	4143.4	3960.1
4	1661.1	1149.2	3637.9	2511
5	611.1	655.5	1274.1	789.3
6	996.9	670.2	1985.9	2049.6
7	Not done			
8	Not done			
9	1499.9	1033.1	3278.7	1952.4
10	2085	1464.6	5749.9	1248.7

Reference values: cystine—43~210 nmol/mg of creatinine, ornithine—20~80 nmol/mg of creatinine, lysine—145~634 nmol/mg of creatinine, and arginine—10~90 nmol/mg of creatinine.

**Table 3 biomedicines-11-02747-t003:** Mutations detected in patients.

Case	Genes	Mutation 1	Mutation 2	Genotypes
1	*SLC3A1*	c.460G>C, p.Ala154Pro ^†^ (Missense)		Hetero A
2	*SLC3A1*	c.1820delT, p.Leu607HisfsTer4 (Frameshift)		Homo A
3	*SLC3A1*	c.1640C>G, p.Ser547Trp (Missense)	c.460G>C, p.Ala154Pro ^†^ (Missense)	Compound hetero A
4	*SLC3A1*	c.1356delG ^†^ (Frameshift)	c.1976A>C, p.Gln659Pro ^†^ (Missense)	Compound hetero A
5	*SLC3A1*	c.647C>T, p.Thr216Met (Missense)	c.2017T>C, p.Cys673Arg (Missense)	Compound hetero A
6	*SLC3A1*	c.1640C>G, p.Ser547Trp (Missense)		Hetero A
7	*SLC7A9*	c.217G>A, p.Gly73Arg (Missense)		Hetero B
8	*SLC7A9*	Exon 12 deletion ^†^ (Deletion)		Hetero B
9	*SLC3A1*	c.256C>T, p.Arg86Trp ^†^ (Missense)	c.405C>G, p.Asn135Lys ^†^ (Missense)	Compound hetero A
10	*SLC3A1*	c.2017T>C, p.Cys673Arg (Missense)		Homo A

^†^ Novel mutations.

**Table 4 biomedicines-11-02747-t004:** Clinical presentation between the type A cystinuria group and the type B cystinuria group.

Clinical Presentation	Type A Cystinuria Group (N = 8)	Type B Cystinuria Group (N = 2)	*p*
Age at onset (years)	15.5 (0.8–26.8)	44.25 (26–62.5)	0.088 ^a^
Sex			0.133 ^b^
Male	6 (75.0%)	0 (0.0%)	
Female	2 (25.0%)	2 (100%)	
Bilateral stones at diagnosis	3 (37.5%)	1 (50%)	0.444 ^b^
Urinary cystine excretion	1550.54 ± 554.17	NA	
Number of interventions	5.5 (1–10)	1.5 (1–2)	0.287 ^a^
Current eGFR	97.0 (69.4–117.2)	84.2 (68.9–99.4)	0.533 ^a^

^a^ Mann–Whitney U test, ^b^ Fisher’s exact test., NA; not available

**Table 5 biomedicines-11-02747-t005:** Previously reported mutational analysis of Korean cystinuria patients.

Reference	Case	Gene	Mutation 1	Mutation 2
Lee et al. (2010) [8]	1	*SLC7A9*	c.535G>A, p.G173R	
Kang et al. (2013) [9]	1	*SLC3A1*	c.1820del, p.L607fs	
Kim et al. (2017) [10]	1	*SLC3A1*	c.647C>T, p.T216M	c.1820delT, p.L607Hfs*4
	2	*SLC3A1*	c.647C>T, p.T216M	c.2017T>C, p.C673R
	3	*SLC3A1*	c.46A>T, p.K16*	c.1500+1G>A, abnormal splicing
	4	*SLC3A1*	c.1820delT, p.L607Hfs*4	c.1820delT, p.L607Hfs*4
	5	*SLC3A1*	c.1820delT, p.L607Hfs*4	c.1976A>C, p.Q659P
	6	*SLC7A9*	c.1445C>T, p.P482L	c.1224+4166_1399+119del4972
	7	*SLC7A9*	c.1224+4166_1399+119del4972	
Lee et al. (2023) [11]	1	*SLC7A9*	c.1305G>A	
	2	*SLC7A9*	c.404A>G	c.988G>A
	3	*SLC3A1*	c.223C>T	c.1318T>C
	4	*SLC7A9*	c.1445C>T	c.1445C>T
	5	*SLC3A1*	c.418G>A	c.1976A>C
	6	*SLC7A9*	c.1060G>A	c.829G>A
	7	*SLC3A1*	c.1500+1G>A	c.1500+1G>A
	8	*SLC3A1*	c.1976A>C	c.2017T>C
	9	*SLC3A1*	c.1820del	1820del

## Data Availability

The data presented in this study are available in the article.

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
