# Peer review of "Clinical Course and Mutational Analysis of Patients with Cystine Stone: A Single-Center Experience"

_biomedicines, 2023, doi:10.3390/biomedicines11102747_

Round 1

Reviewer 1 Report

This interesting study summarizes ten cases of cystinuria a very rare renal disease in Korea . It's depicts clinical complications, medical and surgical therapy as the genetic background. 

Minor comments to improve the manuscript :

Table 1 : please authors give the normal values in healthy persons of the branched amino-acids ;

Table 2 : to what corresponds the acronym PD ?

Discussion section lines 240 to 248, authors please summarize Korean experience of therapy with Captopril and perform a short literature review of cystinuria therapy with it.

Globally good but to be improve by an English or American native

Author Response

­Dear Editor of Biomedicines,

Thank you for your thorough review of our manuscript (biomedicines-2639678) entitled “Clinical course and mutational analysis of patients with cystine stone: A single-center experience” We are also grateful for the opportunity to revise our manuscript. Our manuscript has been carefully revised according to the reviewers’ comments. Please find our responses to the reviewer’s comments starting from the next page.

We hope that our revised paper is acceptable for publication in Biomedicines, and we look forward to receiving your final decision.

Thanks again for your kind consideration.

Sincerely,

Joo Yong Lee, M.D., Ph.D.

Department of Urology, Severance Hospital, Urological Science Institute, Yonsei University College of Medicine, 50-1 Yonsei-ro, Seodaemun-gu, Seoul 03722, Korea

Tel: +82-2-2228-2320; Fax: +82-2-312-2538; E-mail: [email protected]

Reviewer 1

Comment 1-1:

Table 1 : please authors give the normal values in healthy persons of the branched amino-acids ;

Answer 1-1: Thank you for your feedback. We have added the normal value to Table2 as shown below.

9

1499.9

1033.1

3278.7

1952.4

10

2085

1464.6

5749.9

1248.7

Reference values: Cystine 43~210 nmol/mg creatinine, Ornithine 20~80 nmol/mg creatinine, Lysine 145~634 nmol/mg creatinine, Arginine 10~90 nmol/mg creatinine

Comment 1-2:

Table 2 : to what corresponds the acronym PD ?

Answer 1-2: Thanks for your thorough review. The acronym PD corresponds to

pyridoxine (vitamin B6). We’ve added this to the bottom of table1 as shown below.

9

M

38.1

16.0

34.1

Stone

-

+

90

117.2

KC, Capto, PD, Artamin

OpenNL, RIRS

10

F

28.7

26.8

28.3

Stone

+

-

107

114

none

ECIRS, RIRS

HU = hematuria, PU = proteinuria, eGFR = estimated glomerular filtration rate (calculated using the Schwartz formula or CKD-EPI according to age), PD = pyridoxine (vitamin B6), KC = potassium citrate, Capto = captopril, Enala = enalapril, PNL = percutaneous nephrolithotomy, RIRS = retrograde intrarenal surgery, OpenNL = open nephrolithotomy, URSL = ureterorenoscopic lithotomy, ECIRS = endoscopic combined intrarenal surgery, LapaUL= laparoscopic ureterolithotomy. *On the last follow-up imaging test (non-contrast CT or KUB)

Comment 1-3:

Discussion section lines 240 to 248, authors please summarize Korean experience of therapy with Captopril and perform a short literature review of cystinuria therapy with it.

Answer 1-3: Thank you for your comments. Captopril was also prescribed for patients who were unable to take tiopronin, and there were no adverse events associated with its use. We have added the following sentences with references to this section.

“…Captopril, which is a thiol-first generation ACE inhibitor, was also prescribed (Case 1 and Case 9). Although it is currently an off-label administration, there have been pub-lished studies showing effectiveness in cystinuria because captopril has a sufhydryl ligand that bonds with cystine and forms cystine-captopril disulphide, which is more soluble than cystine [27,28]. No significant adverse effects were reported when non-tiopronin drugs were administered, and the addition of tiopronin is being considered based on the availability of the drug and the recurrence of urolithiasis.”

Reviewer 2 Report

General Comments:

The manuscript titled "Genetic Analysis and Clinical Characteristics of Korean Cystinuria Patients" by Jeong et al., provides valuable insights into the genetic and clinical aspects of cystinuria in the Korean population. The authors have presented their findings clearly, and the study is well-structured. However, there are some areas that require attention before the manuscript can be considered for publication. Below are my comments and suggestions:

Major Comments:

1.      Introduction: The introduction provides an overview of cystinuria but lacks a clear statement of the research objectives and the significance of the study. It would be beneficial to state explicitly what the authors aim to achieve through this research and why it is important in the context of cystinuria research.

2.      Methodology: The methodology section is detailed and includes information on patient selection, mutational analysis, and statistical methods. However, it would be helpful to provide more information on the inclusion and exclusion criteria for patient selection. Additionally, the authors should specify the ethical considerations and institutional review board (IRB) approval in more detail.

3.      Results: The results section presents important findings regarding the genetic analysis and clinical characteristics of Korean cystinuria patients. However, the presentation of data could be improved. Consider to polish the tables added.

Minor Comments:

1.      Discussion: While the results are presented well, the discussion section is relatively brief. The authors should expand on the implications of their findings and discuss how their results compare with previous studies in the field. About the calcium based stone formers and the concept of supersaturation also see: https://doi.org/10.3390/cryst11121507 and https://doi.org/10.3390/metabo12030229
Additionally, it would be beneficial to address the limitations of the study more explicitly.

2.      Conclusion: The conclusion section should summarize the key findings and their significance, providing a clear take-home message for the reader.

1.      Language and Grammar: There are some minor grammatical errors and not clear sentence structures throughout the manuscript. I recommend a thorough proofreading for language and style to improve the overall readability of the paper

Author Response

­Dear Editor of Biomedicines,

Thank you for your thorough review of our manuscript (biomedicines-2639678) entitled “Clinical course and mutational analysis of patients with cystine stone: A single-center experience” We are also grateful for the opportunity to revise our manuscript. Our manuscript has been carefully revised according to the reviewers’ comments. Please find our responses to the reviewer’s comments starting from the next page.

We hope that our revised paper is acceptable for publication in Biomedicines, and we look forward to receiving your final decision.

Thanks again for your kind consideration.

Sincerely,

Joo Yong Lee, M.D., Ph.D.

Department of Urology, Severance Hospital, Urological Science Institute, Yonsei University College of Medicine, 50-1 Yonsei-ro, Seodaemun-gu, Seoul 03722, Korea

Tel: +82-2-2228-2320; Fax: +82-2-312-2538; E-mail: [email protected]

Reviewer 2

Comment 2-1-1:

  1. Introduction: The introduction provides an overview of cystinuria but lacks a clear statement of the research objectives and the significance of the study. It would be beneficial to state explicitly what the authors aim to achieve through this research and why it is important in the context of cystinuria research.

Answer 2-1-1: Thank you for your kind feedback. We clearly agree with your comment and have changed the last paragraph of the introduction to make it clear as shown below.

“…As genetic analysis has become increasingly common, various papers have been published including mutational analysis of cystinuria patients, and some genes characteristic of different regions have been reported. However, to date, only four papers have reported on genetic analysis in Koreans, with a total of 18 patients. [8-11]. In this study, we assessed the genetic analysis, phenotype, and clinical courses of 10 novel cystinuria patients which is the largest Korean cohort studied in this manner. Also, we compared the phenotypes of patients with type A and type B cystinuria to determine the association of genotype with the clinical course of cystinuria. Then, we checked the geographical characteristics of Korean cystinuria mutation by adding our data to the existing Korean patients.”

Comment 2-1-2:

  1. Methodology: The methodology section is detailed and includes information on patient selection, mutational analysis, and statistical methods. However, it would be helpful to provide more information on the inclusion and exclusion criteria for patient selection. Additionally, the authors should specify the ethical considerations and institutional review board (IRB) approval in more detail.

Answer 2-1-2: Thank you for your comment. In this study, we did not set any exclusion criteria as we wanted to include all cases with confirmed cystine stones regardless of age or sex. We have added this to the first paragraph of the methodology as follows.

“We searched the databse for stone composition analysis results and selected all cases reported as cystine stones without exception…”

And added the IRB approval date and further explained why informed consent was waived for this study and the measures taken to protect patient privacy as below.

“This study was approved by the Institutional Review Board at Severance Hospital, Yonsei University Health System, Seoul, Korea (Approval No. 4-2023-0008; approval date: Mar 1, 2023). However, the requirement for written informed consent of subjects was waived because this is a retrospective analysis of clinical data obtained while treatment that has already ended and does not involve more than minimal risk to the subject and all patient records and data were anonymized in advance to strictly maintain the confidentiality of the personal information of the subjects throughout the entire research process.”

Comment 2-1-3:

  1. Results: The results section presents important findings regarding the genetic analysis and clinical characteristics of Korean cystinuria patients. However, the presentation of data could be improved. Consider to polish the tables added.

Answer 2-1-3: Thanks for pointing that out. We understand that the placement and size of the tables we previously provided made them difficult to read. Therefore, we have revised the spacing, size, and placement of the tables in this revision to make them more readable, as shown below.

Table 1. Clinical features of 10 patients with cystinuria

Case

Sex

Age

(years)

Onset

(years)

Diagnosis

(years)

Initial symptom

Initial bilateral

stones

Stone

Recurrence*

Initial eGFR

Current eGFR

Medication other than tiopronin

Type of intervention

1

M

38.7

15.0

15.7

Stone

+

+

86

97.7

KC, PD, Capto, Artamin

PNL, RIRS

2

M

17.4

0.9

13.8

UTI

-

+

63.6

70.0

KC, Pyridoxine

OpenNL, URSL

3

M

10.4

0.8

8.8

UTI

-

+

78.0

72.7

KC, PD

ECIRS

4

F

19.7

1.2

13.8

UTI

+

+

111.5

69.4

KC, PD, Enala

PCNL, open NL, ECIRS

5

M

20.6

16.0

16.2

HU, PU

-

+

107.0

108.2

KC, PD

ECIRS

6

M

27.4

18.6

20.8

Stone

-

+

48

96.3

PD

LapaUL

7

F

66.7

62.5

64.0

Stone

-

-

62

68.9

KC, PD

ECIRS

8

F

35.1

26.0

33.2

Stone

+

+

64

99.4

KC, PD, Artamin

ECIRS

9

M

38.1

16.0

34.1

Stone

-

+

90

117.2

KC, Capto, PD, Artamin

OpenNL, RIRS

10

F

28.7

26.8

28.3

Stone

+

-

107

114

none

ECIRS, RIRS

HU = hematuria, PU = proteinuria, eGFR = estimated glomerular filtration rate (calculated using the Schwartz formula or CKD-EPI according to age) (mL/min/1.73m2), PD = pyridoxine (vitamin B6), KC = potassium citrate, Capto = captopril, Enala = enalapril, PNL = percutaneous nephrolithotomy, RIRS = retrograde intrarenal surgery, OpenNL = open nephrolithotomy, URSL = ureterorenoscopic lithotomy, ECIRS = endoscopic combined intrarenal surgery, LapaUL= laparoscopic ureterolithotomy. *On the last follow-up imaging test (non-contrast CT or KUB)

Table 5. Previously reported mutational analysis of Korean cystinuria patients

Reference

Case

Gene

Mutation 1

Mutation 2

Lee et al. (2010) [8]

1

SLC7A9

c.535G>A, p.G173R

Kang et al. (2013) [9]

1

SLC3A1

c.1820del, p.L607fs

Kim et al. (2017) [10]

1

SLC3A1

c.647C>T, p.T216M

c.1820delT, p.L607Hfs*4

2

SLC3A1

c.647C>T, p.T216M

c.2017T>C, p.C673R

3

SLC3A1

c.46A>T, p.K16*

c.1500+1G>A, abnormal splicing

4

SLC3A1

c.1820delT, p.L607Hfs*4

c.1820delT, p.L607Hfs*4

5

SLC3A1

c.1820delT, p.L607Hfs*4

c.1976A>C, p.Q659P

6

SLC7A9

c.1445C>T, p.P482L

c.1224+4166_1399+119del4972

7

SLC7A9

c.1224+4166_1399+119del4972

Lee et al. (2023) [11]

1

SLC7A9

c.1305G>A

2

SLC7A9

c.404A>G

c.988G>A

3

SLC3A1

c.223C>T

c.1318T>C

4

SLC7A9

c.1445C>T

c.1445C>T

5

SLC3A1

c.418G>A

c.1976A>C

6

SLC7A9

c.1060G>A

c.829G>A

7

SLC3A1

c.1500+1G>A

c.1500+1G>A

8

SLC3A1

c.1976A>C

c.2017T>C

9

SLC3A1

c.1820del

1820del

Comment 2-2-1:

  1. Discussion: While the results are presented well, the discussion section is relatively brief. The authors should expand on the implications of their findings and discuss how their results compare with previous studies in the field. About the calcium based stone formers and the concept of supersaturation also see: https://doi.org/10.3390/cryst11121507 and https://doi.org/10.3390/metabo12030229

Additionally, it would be beneficial to address the limitations of the study more explicitly.

Answer 2-2-1: Thank you for the suggestion. In the original Discussion section, we intended to compare our findings with those of previously reported studies, but upon further review, we realized that the organization of the paragraphs didn't make that clear enough, and we compromised the overall organization. Also, we also added that although the association between genotype and clinical course was not clear from genetic analysis, genetic analysis still plays an important role in understanding the pathogenesis of cystinuria. Lastly, we presented the limitations of this study more clearly in the last paragraph of the discussion.

“The genotype and phenotype of the largest cohort of cystinuria patients from a single institution in Korea are summarized in this study. Six novel mutations were identified and considered to be pathogenic based on their frequency in the normal population and the expected high pathogenicity of in silico analysis.

The clinical presentation of cystinuria varies in the literature. For example, the male-to-female ratio is reported to be 2:1 which differs from the previously reported 1:3 ra-tio in a Korean cohort [6,10]. Dello Strologo et al. reported that male patients had a more severe phenotype than female patients [14]; However, our data showed a relatively more repetitive intervention rate in female patients (median, male=3.5 vs female=4.0). Although there is variation in the literature, less than 80% of patients reportedly have their first stones within the first two decades of life, and 1–16% of patients present with the first stone before age 40, which closely recapitulates the results in our cohort [15,16].

Unlike calcium-based stone formers, for which up to 25% reportedly have bilateral stone formation, cystinuria is characterized by up to 75% bilateral stone formation. How-ever, no significant difference in renal function between bilateral and unilateral stone formers has been reported, even though bilateral stone formers require relatively more intervention than unilateral stone formers [17]. In this study, four patients had bilateral stones at the time of initial intervention and these patients had more additional intervention than those who had unilateral stones (5.5±2.4 vs 3.3±3.6, P=0.326). Nevertheless, current renal function with eGFR was similar between the two groups (95.1±18.6 vs 88.9±21.2, P=0.646).

Regarding the genotypes of cystinuria, type A cystinuria was generally considered as an autosomal recessive disease; however, subsequent studies have identified cases of high urinary cystine excretion and stone formation in heterozygous individuals [3,18]. In this study, two patients with heterozygous type A mutation showed above-normal cystinuria levels. The type B mutation is known to be autosomal dominant with incomplete dominance. Consistent with this knowledge, two type B patients in our cohort showed a heterozygous genotype without familial history.

Of the 12 mutations reported, the most common were missense mutations, followed by two frameshifts, and one deletion. Wong et al. hypothesized that patients with mis-sense mutations have less severe disease because missense mutations result in a change to a single amino acid within the protein compared with all other types of mutations that cause larger genomic alterations [19]. In their study, patients with at least one missense mutation in SLC3A1 had significantly lower levels of urinary lysine, arginine, and ornithine but not cystine than patients with any other combination of mutations. In contrast, our data showed no significant difference in amino acid excretion levels in patients with missense mutations (cystine, P=0.945; ornithine, P=0.402; lysine, P=0.991; and arginine, P=0.537).

We compared the clinical presentations between type A and type B cystinuria and observed no significant differences. These results are consistent with previous studies [10,14,19,20]. Therefore, current research suggests that there is no difference in the clinical course and characteristics of cystinuria based on genotype, and both types require ongoing renal function testing and continued follow-up for stone recurrence. However, these results do not mean that genetic testing for cystinuria is not necessary. For example, studies have been published showing no pathogenic variants in SLC3A1 or SLC7A9 in 5~10% of patients with confirmed cystine stones [21,22]. Wu et al. reported discrepancy between the clinical and genetic prevalence (1 in 7,000 individuals vs 1 in 30,585 estimates) and suggested novel causal genes, undiscovered pathogenic variants, alternative inheritance models, founder effect, epigenetic modification, environmental factors, modifier genes, or other modification factors as possible causes [23]. Given these considerations, genetic analysis still has an important role to understand the pathogenesis of cystinuria.

Geographical variations in cystinuria mutations have been reported. For example, p.T216M in SLC3A1 and p.G105R in SLC7A9 are common in Greece and dupE5E9 is more commonly found in Germany. In the Asian population, SLC7A9 c.1445C > T (p. P482L) is common and might indicate a mutation hotspot in the Japanese population [6,24].  Among the total of 24 identified mutations, the aforementioned common mutations p.T216M and p.P482L were also found. However, the common mutations in the Korean population was c.1820delT, c.1976A>C, c.2017T>C in SLC3A1 (28.5% in total).

Although there is an increased risk of CKD in patients with cystinuria, actual kidney failure is known to be less than 5% [25], and our patients presented with at least CKD stage 2 at their last follow-up evaluation.

Prot-Bertoye et al. showed that CKD progression in cystinuria patients is associated with age, hypertension, and severe damage of the renal parenchyma [26]. Therefore, one important component of treating cystinuria is to prevent an increase in stone burden, thereby reducing the number of urological interventions and damage to renal parenchyma [27,28]. At our institution, tiopronin was prescribed first if the patient had a high stone burden and frequent recurrences. Unfortunately, due to production and supply shortages, only three patients are currently being treated with tiopronin (Cases 5, 7, and 9). Of the pa-tients who attempted tiopronin therapy, two patients discontinued use due to adverse drug reactions. Case 4 developed OT/PT elevations and Case 6 discontinued use due to allergic symptoms including fever, colitis, and presumed drug eruption. Two additional patients (Case 1 and Case 3) had no adverse events but were discontinued because the drug was no longer available and are on a plan to receive it when it becomes available. If medication was needed and tiopronin was not available, medications such as potassium citrate or D-penicillamine (Case 1 and Case 8) were considered [29]. Captopril, which is a thiol-first generation ACE inhibitor, was also prescribed (Case 1 and Case 9). Although it is currently an off-label administration, there have been published studies showing effectiveness in cystinuria because captopril has a sulfhydryl ligand that bonds with cystine and forms cystine-captopril disulfide, which is more soluble than cystine [30,31]. No significant adverse effects were reported when non-tiopronin drugs were administered, and the addition of tiopronin is being considered based on the availability of the drug and the recurrence of urolithiasis.

This study had some limitations. First, the 10 patients in this study is a relatively small sample size. Only two type B mutations were identified in this cohort, and these two patients were not available for urinary amino acid excretion testing. More patients should be enrolled in future studies to more precisely compare the clinical course according to mutation. Second, the study relied only on patient statements to investigate family history, and did not obtain genetic testing or laboratory data from family members. Such results would have provided additional information on the inheritance pattern and clinical course of cystinuria.”

Comment 2-2-2:

  1. Conclusion: The conclusion section should summarize the key findings and their significance, providing a clear take-home message for the reader.

Answer 2-2-2: Thank you for your feedback. We’ve changed the paragraph to summarize the findings and provide more clear view as below.

“We investigated genotypes and clinical courses of 10 Korean patients with cystinuria who were not previously reported. No significant association has been identified between type A and type B cystinuria on clinical course, including recurrence of stones. Mutations in SLC3A1 gene were more frequently observed in Korean population.”

Round 2

Reviewer 2 Report

The manuscript has been improved. Nevertheless, reconsider this previous recommendation:

Discussion: While the results are presented well, the discussion section is relatively brief. The authors should expand on the implications of their findings and discuss how their results compare with previous studies in the field. About the calcium based stone formers and the concept of supersaturation also see: https://doi.org/10.3390/cryst11121507 and https://doi.org/10.3390/metabo12030229

-

Author Response

­Dear Editor of Biomedicines,

Thank you for your thorough review of our manuscript (biomedicines-2639678) entitled “Clinical course and mutational analysis of patients with cystine stone: A single-center experience” We are also grateful for the opportunity to revise our manuscript. Our manuscript has been carefully revised according to the reviewers’ comments. Please find our responses to the reviewer’s comments starting from the next page.

We hope that our revised paper is acceptable for publication in Biomedicines, and we look forward to receiving your final decision.

Thanks again for your kind consideration.

Sincerely,

Keum Hwa Lee, M.D.

Department of Pediatrics, Yonsei University College of Medicine,

50-1 Yonsei-ro, Seodaemun-gu, Seoul 03722, Korea

Tel: +82-2-2228-2050 / Fax: + 82-2-393-9118 /E-mail: [email protected]  

Joo Yong Lee, M.D., Ph.D.

Department of Urology, Severance Hospital,

Urological Science Institute, Yonsei University College of Medicine,

50-1 Yonsei-ro, Seodaemun-gu, Seoul 03722, Korea

Tel: +82-2-2228-2320; Fax: +82-2-312-2538; E-mail: [email protected]

Reviewer 2

Comment 2-1:

Discussion: While the results are presented well, the discussion section is relatively brief. The authors should expand on the implications of their findings and discuss how their results compare with previous studies in the field. About the calcium based stone formers and the concept of supersaturation also see: https://doi.org/10.3390/cryst11121507 and https://doi.org/10.3390/metabo12030229

Answer 2-1:

Thank you for your kind feedback and inspiring references. Upon further review, we have also added a comparison between the clinical presentations of previously reported cystinuria patients and the results of our study. Additionally, we emphasized that the inheritance pattern confirmed through genetic testing can serve as a basis for the necessity of ongoing mutational analysis. Furthermore, we gained valuable ideas for future cystinuria research from the references you provided and also included this content as below.

“The genotype and phenotype of the largest cohort of cystinuria patients from a single institution in Korea are summarized in this study. Six novel mutations were identified and considered to be pathogenic based on their frequency in the normal population and the expected high pathogenicity of in silico analysis.

The clinical presentation of cystinuria varies in the literature. For example, the male-to-female ratio is reported to be 3:2 which differs from the previously reported Kore-an cohorts [10,11]. Dello Strologo et al. reported that male patients had a more severe phenotype than female patients [14]; However, our data showed a relatively more repetitive intervention rate in female patients (median, male=3.5 vs female=4.0). Although there is variation in the literature, less than 80% of patients reportedly have their first stones within the first two decades of life, and 1–16% of patients present with the first stone be-fore age 40, which closely recapitulates the results in our cohort [15,16].

Unlike calcium-based stone formers, for which up to 25% reportedly have bilateral stone formation, cystinuria is characterized by up to 75% bilateral stone formation. However, no significant difference in renal function between bilateral and unilateral stone formers has been reported, even though bilateral stone formers require relatively more intervention than unilateral stone formers [17]. In this study, four patients had bilateral stones at the time of initial intervention and these patients had more additional intervention than those who had unilateral stones (5.5±2.4 vs 3.3±3.6, P=0.326). Nevertheless, current renal function with eGFR was similar between the two groups (95.1±18.6 vs 88.9±21.2, P=0.646).

Although there is an increased risk of CKD in patients with cystinuria, actual kidney failure is known to be less than 5% [18], and our patients presented with at least CKD stage 2 at their last follow-up evaluation.

Prot-Bertoye et al. showed that CKD progression in cystinuria patients is associated with age, hypertension, and severe damage of the renal parenchyma [19]. Therefore, one important component of treating cystinuria is to prevent an increase in stone burden, thereby reducing the number of urological interventions and damage to renal parenchyma [20,21]. At our institution, tiopronin was prescribed first if the patient had a high stone burden and frequent recurrences. Unfortunately, due to production and supply shortages, only three patients are currently being treated with tiopronin (Cases 5, 7, and 9). Of the patients who attempted tiopronin therapy, two patients discontinued use due to adverse drug reactions. Case 4 developed OT/PT elevations and Case 6 discontinued use due to allergic symptoms including fever, colitis, and presumed drug eruption. Two additional patients (Case 1 and Case 3) had no adverse events but were discontinued because the drug was no longer available and are on a plan to receive it when it becomes available. If medication was needed and tiopronin was not available, medications such as potassium citrate or D-penicillamine (Case 1 and Case 8) were considered [22]. Captopril, which is a thiol-first generation ACE inhibitor, was also prescribed (Case 1 and Case 9). Although it is currently an off-label administration, there have been published studies showing effectiveness in cystinuria because captopril has a sulfhydryl ligand that bonds with cystine and forms cystine-captopril disulfide, which is more soluble than cystine [23,24]. No significant adverse effects were reported when non-tiopronin drugs were administered, and the addition of tiopronin is being considered based on the availability of the drug and the recurrence of urolithiasis.

Regarding the genotypes of cystinuria, type A cystinuria was generally considered as an autosomal recessive disease; however, subsequent studies have identified cases of high urinary cystine excretion and stone formation in heterozygous individuals [3,25]. In this study, two patients with heterozygous type A mutation showed above-normal cystinuria levels. The type B mutation is known to be autosomal dominant with incomplete penetrance. Consistent with this knowledge, two type B patients in our cohort showed a heterozygous genotype without familial history.

Of the 12 mutations reported, the most common were missense mutations, followed by two frameshifts, and one deletion. Wong et al. hypothesized that patients with mis-sense mutations have less severe disease because missense mutations result in a change to a single amino acid within the protein compared with all other types of mutations that cause larger genomic alterations [26]. In their study, patients with at least one missense mutation in SLC3A1 had significantly lower levels of urinary lysine, arginine, and ornithine but not cystine than patients with any other combination of mutations. In contrast, our data showed no significant difference in amino acid excretion levels in patients with missense mutations (cystine, P=0.945; ornithine, P=0.402; lysine, P=0.991; and arginine, P=0.537).

Geographical variations in cystinuria mutations have been reported. For example, p.T216M in SLC3A1 and p.G105R in SLC7A9 are common in Greece and dupE5E9 is more commonly found in Germany. In the Asian population, SLC7A9 c.1445C > T (p. P482L) is common and might indicate a mutation hotspot in the Japanese population [6,27,28].  Among the total of 24 identified mutations, the aforementioned common mutations p.T216M and p.P482L were also found. However, the common mutations in the Korean population was c.1820delT, c.1976A>C, c.2017T>C in SLC3A1 (28.5% in total).

We compared the clinical presentations between type A and type B cystinuria and observed no significant differences. These results are consistent with previous studies [10,14,26,29]. Therefore, current research suggests that there is no difference in the clinical course and characteristics of cystinuria based on genotype, and both types require ongoing renal function testing and continued follow-up for stone recurrence. However, these results do not mean that genetic testing for cystinuria is not necessary. For example, studies have been published showing no pathogenic variants in SLC3A1 or SLC7A9 in 5~10% of patients with confirmed cystine stones [30,31]. Wu et al. reported discrepancy between the clinical and genetic prevalence (1 in 7,000 individuals vs 1 in 30,585 estimates) and suggested novel causal genes, undiscovered pathogenic variants, alternative inheritance models, founder effect, epigenetic modification, environmental factors, modifier genes, or other modification factors as possible causes [32]. In our cohort, family history was identified in only 2 out of 10 patients, and two patients (Case 1 and Case 6) showed a heterozygous SLC3A1 mutation with no family history, indicating autosomal dominant with in-complete penetrance rather than autosomal recessive. In this regard, previous studies suggested existence of genetic modifiers or novel cystine transporter [33,34]. Given these considerations, genetic analysis still has an important role to understand the pathogenesis of cystinuria.

Furthermore, while research on cystinuria has primarily focused on clinical and genetic aspects to date, considering patient groups without SLC3A1/SLC7A9 mutations, it appears that additional studies are necessary, including research on the physicochemical factors related to cystine aggregation, similar to that observed in calcium stone studies, and investigations into other urinary tract symptoms that may arise due to cystine crystal formation [35,36].

Summarizing, this study followed the clinical course of 10 Korean patients with cystinuria, and despite recurrent stones, the patients' renal function was well maintained with continuous follow-up. Although no statistical association between genotype and phenotype was found, mutation analysis showed that SLC3A1 mutation was more frequent in Korean patients, in contrast to previous analyses in Europeans and Japanese.

This study had some limitations. First, the 10 patients in this study is a relatively small sample size. Only two type B mutations were identified in this cohort, and these two patients were not available for urinary amino acid excretion testing. More patients should be enrolled in future studies to more precisely compare the clinical course according to mutation. Second, the study relied only on patient statements to investigate family history, and did not obtain genetic testing or laboratory data from family members. Such results would have provided additional information on the inheritance pattern and clinical course of cystinuria.”

References

  1. Nagamori, S.; Wiriyasermkul, P.; Guarch, M.E.; Okuyama, H.; Nakagomi, S.; Tadagaki, K.; Nishinaka, Y.; Bodoy, S.; Takafuji, K.; Okuda, S.; et al. Novel cystine transporter in renal proximal tubule identified as a missing partner of cystinuria-related plasma membrane protein rBAT/SLC3A1. Proc Natl Acad Sci U S A 2016, 113, 775-780, doi:10.1073/pnas.1519959113.
  2. Sasaki, H.; Sasaki, T.; Hiura, K.; Watanabe, M.; Sasaki, N. A mouse model of type B cystinuria due to spontaneous mutation in FVB/NJcl mice. Urolithiasis 2022, 50, 679-684, doi:10.1007/s00240-022-01356-9.
  3. Rossi, M.; Barone, B.; Di Domenico, D.; Esposito, R.; Fabozzi, A.; D’Errico, G.; Prezioso, D. Correlation between Ion Composition of Oligomineral Water and Calcium Oxalate Crystal Formation. Crystals 2021, 11, 1507.
  4. Matsuo, T.; Ito, H.; Mitsunari, K.; Ohba, K.; Miyata, Y. Relationship between Urinary Calcium Excretion and Lower Urinary Tract Symptoms. Metabolites 2022, 12, 229.